# Correlation between Salivary Levels of IGF-1, IGFBP-3, IGF-1/IGFBP3 Ratio with Skeletal Maturity Using Hand-Wrist Radiographs

**DOI:** 10.3390/ijerph19063723

**Published:** 2022-03-21

**Authors:** Abdullah Almalki, Julie Toby Thomas, Abdul Rehman Ahmed Khan, Basim Almulhim, Abdullah Alassaf, Sara Ayid Alghamdi, Betsy Joseph, Ali Alqerban, Saud Alotaibi

**Affiliations:** 1Department of Preventive Dental Sciences (Orthodontics), College of Dentistry, Majmaah University, Al-Majmaah 11952, Saudi Arabia; ae.almalki@mu.edu.sa (A.A.); ab.khan@mu.edu.sa (A.R.A.K.); alotaibi.s@mu.edu.sa (S.A.); 2Department of Preventive Dental Sciences (Periodontics), College of Dentistry, Majmaah University, Al-Majmaah 11952, Saudi Arabia; 3Department of Preventive Dental Sciences (Pedodontics), College of Dentistry, Majmaah University, Al-Majmaah 11952, Saudi Arabia; b.almulhim@mu.edu.sa (B.A.); am.assaf@mu.edu.sa (A.A.); sa.mohammed@mu.edu.sa (S.A.A.); 4Department of Periodontics, Saveetha Institute of Medical Sciences, Chennai 600077, India; betsyj.sdc@saveetha.com; 5Department of Preventive Dental Sciences (Orthodontics), College of Dentistry, Prince Sattam Bin Abdulaziz University, Al-kharj 11942, Saudi Arabia; a.alqerban@dau.edu.sa; 6Department of Preventive Dental Sciences, College of Dentistry, Dar al Uloom University, Riyadh 45142, Saudi Arabia

**Keywords:** orthodontics, insulin-like growth factor, insulin-like growth factor binding protein, skeletal maturity, salivary diagnostics, hand wrist radiographs

## Abstract

Objective: The relevance of growth determination in orthodontics is driving the search for the most precise and least invasive way of tracking the pubertal growth spurt. Our aim was to explore whether minimally invasive salivary estimation of biomarkers Insulin-like growth factor (IGF-1) and Insulin-like growth factor binding protein-3 (IGFBP-3) could be used to estimate skeletal maturity for clinical convenience, especially in children and adolescent age groups. Materials and Method: The cross-sectional study was conducted on 90 participants (56 girls and 34 males) with ages ranging from 6 to 25 years. Each subject’s hand-wrist radiograph was categorized based on skeletal maturity, and saliva samples were estimated for IGF-1 and IGFBP-3 using the respective ELISA kits. Kruskal–Wallis nonparametric ANOVA was applied to compare different skeletal stages. Results: The study demonstrated low salivary IGF-1 levels at the prepubertal stage, with increase during pubertal onset and peak pubertal stage followed by a decline during pubertal deceleration to growth completion. Spearman’s correlation coefficient demonstrated a strong positive association (r = 0.98 *p* < 0.01) between salivary IGF/IGFBP-3 ratio and different stages of skeletal maturity. Conclusion: Salivary IGF-1, IGFBP-3, and IGF/IGFBP-3 ratio could serve as a potential biochemical marker for predicting the completion of skeletal maturity.

## 1. Introduction

Success in orthodontic therapy depends on precise and accurate decision-making based on the skeletal maturity of individuals. The importance of growth determination in orthodontics is the driving force in searching for the most accurate and least invasive method of tracking the pubertal growth spurt, which is considered a beneficial period for orthopedic (skeletal) corrections related to the craniofacial complex [1]. Various indicators have been adopted to estimate the skeletal maturity of individuals, including determination of chronological age [2], peak height velocity [3], dental age [4], somatic and sexual maturity [5], and assessment of hand-wrist radiographs and lateral cephalograms [6,7,8,9]. The skeletal age of a child indicates his/her level of biological and structural maturity better than the chronological age.

The use of indicators related to cervical maturation has a comparative advantage of avoiding additional radiographic exposure. Nevertheless, the morphological differences observed during quantification of cervical vertebral maturation (QCVM) indicators are not as pronounced and identifiable as in the classical hand-wrist method. In addition, the tendency to distort and magnify the lateral cephalograms taken for QCVM may affect the comparability and validity while interpreting linear measurements and the shape of the vertebrae [10]. Therefore, using cervical vertebrae as an indicator of maturity is highly subjective and lacks the simplicity of evaluation compared to the hand wrist technique [11]. 

Recent studies have focused on the diagnostic efficacy of serum biomarkers such as growth hormone (GH), insulin-like growth factor-1 (IGF-1), insulin growth factor binding protein (IGFBP-3), osteocalcin, dehydroepiandrosterone (DHEA), cortisol, and alkaline phosphatase using enzyme immunoassays, radioimmunoassays, and immunoradiometric assays to predict growth maturity with higher accuracy and precision [12]. 

IGF-I, known as somatomedin C, is a mediator of growth hormone (GH) function and plays an essential role in postnatal longitudinal bone growth. In addition, IGF-1 levels do not fluctuate diurnally, such as GH, making it a useful diagnostic tool for determining GH status [1]. Many studies have been conducted correlating IGF-1 levels in blood serum with various skeletal maturity stages, but a collection of serum samples remains an invasive procedure [10,13,14]. IGFs are generally bound to six well-characterized insulin growth factor binding proteins (IGFBP1-6), designated IGFBP-1 to -6 in biological fluids such as blood serum, gingival crevicular fluid (GCF), or saliva [15]. As cited by Toia M et al. in 2005, IGBF-3 is the most abundant IGFBP and regulates the amount of free, bioactive IGF-1 in circulation. In addition, IGFBPs are directly associated with a variety of extracellular and cell surface molecules, ensuring their corresponding effects on critical biological processes such as the modulation of bone cell proliferation [16]. 

Limited studies have been reported thus far in the literature related to IGF-1/IGFBP-3 ratio levels in saliva. In 2006, Henkin RI and Velicu I were the first to discover IGF-1 and IGFBP-3 in human saliva and nasal mucus. They established that IGF-1 and IGFBP-3 were 5–10 times higher in nasal mucus than in saliva [17]. Salivary estimation of these biomarkers can be a minimally invasive potential tool for estimating skeletal maturity for clinical convenience, especially in children and adolescent age groups. Hence, the objective of this study was to estimate and compare the salivary levels of IGF-1 and IGFBP-3 in whole unstimulated saliva at various skeletal maturity stages based on skeletal maturity indicators in hand wrist radiographs and to correlate the IGF-1/IGFBP-3 molar ratio with different skeletal ages. 

## 2. Materials and Methods

### 2.1. Ethics

This cross-sectional study was undertaken in the College of Dentistry, Majmaah University, Zulfi, Saudi Arabia, between 10 January 2021 to 10 April 2021 after approval by the institutional ethical committee of Majmaah University, Saudi Arabia (Research Number: MUREC Nov.08/COM-2020/8-2; dated 8 November 2020) in compliance with the Helsinki Declaration. 

### 2.2. Population

The sample size (n) was calculated by the formula
(1)$$n=\frac{2S^{2}}{d^{2}}{\left(Z_{\alpha /2}+Z_{1-\beta }\right)}^{2}$$
where *Z_α/2_* = 1.96 and *Z_1−β_* = 1.28 are the 95% confidence values obtained from a standard normal distribution. At least 84 subjects were required to detect a significant difference in IGF-1 of effect size (*d*) 0.4 and standard deviation (*S*) of 0.8 obtained from a pilot survey using 15 patients. To tackle drop-out or non-response, 5% of *n* was added to obtain the final sample size. Hence, the final sample size was fixed at 90 after considering the time and cost of the investigation.

A convenience sampling technique selected an estimated sample size of 90 subjects (56 females and 34 males) from the outpatient department of Preventive Dental Science. Prior to reporting at the reception desk, all the patients were screened for COVID-19 by a well-trained dental nurse at the triage station located at the entrance of the dental hospital. Details pertaining to the history of travel within the past 6 months, history of exposure to infected persons, and presence of symptoms such as fever, cold, cough, breathing difficulty, muscle pain, and headache for the past 14 days were recorded in the triage form [18,19]. Apart from the screening procedures, it was mandatory for all patients to provide the updated version of the health app provided by the Ministry of Health in Saudi Arabia. 

Systemically healthy subjects who had come for an orthodontic consultation before treatment and were willing to participate in the study with age criteria ranging between 6 to 25 years were included in the study. Exclusion criteria for the study included subjects with chronic systemic diseases; diseases affecting growth, such as vitamin D, parathyroid, growth, and thyroid hormone disorders; renal dysfunction; diabetes mellitus; growth abnormalities; bleeding disorders; those taking medications affecting bone metabolism in the past six months; xerostomia; history of COVID-19 infection within the past 3 months, and those who had undergone previous fixed or functional orthodontic treatment. All participants and parents of subjects under 14 years of age signed an informed consent form before being enrolled.

### 2.3. Data Collection

General examinations, including height, weight, previous medical status, medical history of COVID-19, previous dental status, and family history; and clinical parameters, such as simplified oral hygiene index (OHI-S), bleeding on probing (BOP), probing pocket depth, clinical attachment loss (CAL), and community periodontal index (CPI), to monitor periodontal status were recorded electronically by a single examiner [20,21,22,23]. 

Radiographs of left-hand wrists were taken for all subjects with exposure at 80 kVp and 9 mA for 1.25 s with the same digital radiographic unit. The images were taken by Kodak 8000 (Carestream Health, Inc., Rochester, NY, USA) panoramic and cephalometric system was analyzed with Apteryx Imaging software (Apteryx Imaging Inc. Vancouver, BC, Canada). Two investigators who were blinded to the patient’s personal details, clinical assessment recordings, and analysis results of salivary biomarkers made the radiographic interpretation. Furthermore, these radiographs were segregated into 5 different stages of skeletal maturity based on ossification of the ulnar sesamoid of the first finger, middle and distal phalanges of the third finger, and the epiphysis of the radius as stated by Hagg and Taranger [9]. Intra- and inter-investigator reliability was estimated to be 0.89 and 0.83, respectively, using the kappa coefficient.

### 2.4. Salivary Collection 

Five milliliters of unstimulated saliva was collected in a sterile graduated leak-proof Eppendorf tube with a screw cap by the spitting method [18,24]. To avoid diurnal variation, all samples were collected between 9 and 10 am after the patients had thoroughly rinsed their mouths with water for 30 s. Subjects were instructed to collect saliva on the floor of the mouth after being seated in an upright position and subsequently spit it out into the tube every 60 s. The saliva samples were sealed, labeled for identification by the third investigator, and stored at −80 °C in the laboratory of Clinical Biochemistry in the Basic Science Department of the College of Dentistry, Majmaah University, until further analysis.

### 2.5. Quantitative Assessment of Salivary IGF-1 and IGFBP-3 Using Enzyme-Linked Immunosorbent Assay (ELISA) 

Collected samples were analyzed for IGF-1 and IGFBP-3 by enzyme-linked immune sorbent assay (ELISA) within 4 months. The stored saliva was thawed and centrifuged at 3500 RPM for 10 min to remove particles. The clear supernatant was collected to estimate IGF-1 and IGFBP3. Commercially available ELISA kits procured from Cloud-Clone Corp Katy, TX, USA for IGF1 (Product No. SEA050Hu) and IGFBP3 (Product No. SEA054Hu) were used according to the manufacturer’s instructions. 

Two hundred microliters of the standard solution and obtained supernatant solution were pipetted into each of the wells of the ELISA plates pre-coated with the antibodies for each of the biochemical constituents. The plates were incubated at 37 °C for 1 h. The clear liquid was removed from each well. Then, 200 µL of “prepared detection reagent A” from the kit was added. This solution was mixed, incubated at 37 °C for 1 hour, further aspirated and washed 3 times. Later, 200 µL of prepared detection reagent B was added, incubated for 30 min at 37 °C, then aspirated and washed 5 times. To this, 180 µL of substrate solution was added and incubated for 10–20 min at 37 °C. Finally, 100 µL of stop solution was added, and the optical density (OD) was immediately read using an ELISA reader at 450 nm. The OD value of the standard (*X*-axis) was plotted against the log of the concentration of the standard (*Y*-axis). The best-fit straight line was drawn through the standard points, as determined by regression analysis, using plotting software curve expert version 2.6.4. The concentration of the biochemical constituents in the saliva samples was determined from this curve. The Detection range of IGF-1 was 0.156–10 ng/mL and that of IGFBP-3 was 1.56–100 ng/mL. Sensitivity of IGF-1: −0.067 ng/mL and IGFBP3: −0.61 ng/mL.

### 2.6. Statistical Analysis

The Kolmogorov–Smirnov test showed that the data distribution of IGF-1 and IGFBP-3 violated the normality assumption, and we opted for nonparametric statistical methods to analyze the collected data on study variables. Kruskal–Wallis nonparametric ANOVA was applied to compare different skeletal stages as determined by the hand-wrist radiograph (stages 1–5) followed by the Mann–Whitney U test for pairwise comparisons between stages. Bonferroni correction was carried out, and the alpha value was set at 0.005. The Spearman rank correlation coefficient was used to examine the salivary IGF1/IGFBP3 ratios at different stages of skeletal development. Since nonparametric methods were employed, the summary statistics used were median, range, frequency, and percentage. A calculated *p*-value less than 0.05 was considered statistically significant. All analyses were carried out with the help of the commercially available statistical package SPSS v.23 (IBM Corporation, Armonk, NY, USA) for Windows.

## 3. Results

A total of 90 patients participated in this study. Figure 1 shows examples of hand-wrist radiographs depicting different skeletal maturity stages. 

Table 1 shows the frequency and percentage distribution of demographic details and clinical parameters to assess the periodontal status of the participants included in the study. The majority of male patients were older than 20 years of age (41.2%), while most female patients belonged to the 14–20 years of age group (38.9%). The majority of the patients in the sample belonged to rural areas (66.7%). The majority of patients did not have a family history of periodontal disease (81.1%). The frequency table shows that a majority of patients brushed only once daily (58.9%). The majority of the patients reported a good level of OHI-S (57.8%). Most of them also did not show any BOP (52.2%), normal probing pocket depth (80%), a CPI score of 0 (51.1%), and no CAL.

Table 2 clearly shows a significant interaction effect of stages with gender on the average levels of IGF-1 and IGFBP-3. The Kruskal–Wallis test showed that the average level of IGF-1 did not differ significantly (*p* > 0.05) among different skeletal stages as determined by the hand-wrist radiograph in male patients (stages 1–5), while for female patients, it differed significantly (*p* < 0.01). The highest IGF-1 in female patients was reported in stage 4 (2.32), followed by stage 3 (2.22), and the lowest value was reported in stage 1 (0.57). In male and female patients, a significant stagewise difference (*p* < 0.01) in the average level of IGFBP-3 was observed. The highest average IGFBP3 was reported in male patients for stage 3 (4.29), followed by stage 5 (4.25), and for female patients in stage 3 (4.31) and stage 4 (4.06). 

The IGF-1 levels of female patients differed significantly (*p* < 0.001) between (pairs) stage 1 with stage 3, stage 1 with stage 4, stage 1 and 5, and stage 3 with stage 5. IGFBP-3 levels of male patients differed significantly (*p* < 0.001) between pairs stage 1 with stage 3, stage 3 and stage 4, stage 4 and 5. Statistically significant differences (*p* < 0.005) were observed in IGFBP-3 levels among female patients (between stages 1 and stage 3; stage 2 and stage 3; stage 3 and 5.

The Mann–Whitney test showed no significant mean differences (*p* > 0.05) between male and female patients based on IGF-1 in the intrastage comparisons of stages 1, 3, 4, and 5. In stage 2, the male patients reported a significantly (*p* < 0.05) higher level of average IGF-1 (2.52) than the female patients (0.87). However, for stages 1 and 5, gender-wise differences were noted (*p* < 0.01) in the average level of IGFBP-3. In both cases, females reported comparatively lower values (3.23 and 3.36) than their counterparts.

Spearman’s correlation coefficient between the salivary IGF/IGFBP3 ratios at different stages of skeletal development as assessed by the hand-wrist radiograph reported a significantly high level of positive association (r = 0.98. *p* < 0.01). The inference is that as the IGF-1/IGFBP-3 ratio increases, the pubertal peak becomes significant (Figure 2). 

## 4. Discussion

Determination of skeletal maturity plays a significant role in orthodontic diagnosis and treatment planning. It helps clinicians anticipate mandibular growth patterns and assess bone maturity. Studies related to growth patterns and early maturity in children have demonstrated variation among the sexes [25]. Several radiographic methods have been used to predict the timing of pubertal growth spurt as part of routine diagnostics, but repeated radiation exposure raises ethical concerns. Therefore, in this study, human ELISA kits were used to estimate the salivary biological markers IGF-1 and IGFBP-3 to determine the degree of bone maturation.

All participants included in this study were devoid of any systemic comorbidities, such as diabetes, renal disease, hypertension, and diseases affecting bone hemostasis, such as disorders in vitamin D, parathyroid, and growth hormone. They were further grouped based on the indicators of skeletal maturity using hand-wrist radiographs [9]. The reliability of hand-wrist radiographs to predict the overall skeletal growth rate has been well established and demonstrated to be accurate for various racial groups [26]. Conflicting studies have pointed to certain limitations, such as polymorphisms in the ossification sequence and skeletal sexual dimorphism in wrist bones [27]. The validity of hand-wrist skeletal maturity indicators to predict craniofacial growth has been quite controversial, as the bones of the craniofacial skeleton are formed by intramembranous ossification, unlike the other bones of the body, which are developed by endochondral ossification. Functional areas in the craniofacial region show different growth responses to various systemic and local environmental conditions [28]. Sato et al. recommended that the prediction of pubertal growth is more accurate when morphologic, biologic, or genetic indicators are used to supplement radiologic evaluation. In addition, ethical concerns related to repeated radiography have prioritized research on biological markers to assess growth maturity [29].

IGF-1 (Somatomedin C), which is primarily synthesized in the liver in response to growth hormone, is critical for human bone growth and development. Apart from identifying growth disorders, estimation of IGF-1 could also predict skeletal development in puberty to achieve favorable outcomes in orthodontic and orthognathic interventions. This study demonstrated low salivary IGF-1 levels of 0.93 ng/mL at the prepubertal stage. The levels varied tremendously in pubertal onset, with a mean of 2.52 ng/mL in males and a peak pubertal stage to a mean of 2.58 ng/mL. There was a decline in the levels of IGF-1 in the stage of pubertal deceleration to growth completion (1.67–1.09 ng/mL). Salivary IGF-1 levels demonstrated a gradual rise from stage 1 to the onset of puberty, with a sharp rise at peak pubertal stage to pubertal deceleration (2.22–2.32 ng/mL). The results obtained on comparing the salivary IGF-1 levels among females belonging to different age groups were statistically significant (*p* = 0.004). The Pearson correlation coefficient demonstrated a strong positive correlation (r = 0.6) from stage 1 to stage 4. This study’s findings were inconsistent with the study done by Masoud et al. in 2009, where IGF-1 levels in blood spot samples were correlated with different skeletal maturation indicators in hand-wrist radiographs [13]. The levels were significantly higher in pubertal stages than in prepubertal and postpubertal stages. As age advanced, it was reported that the IGF-1 levels decreased gradually in the postpubertal group [25,30].

Analysis of IGF-1 levels in blood samples by Sinha et al. in 2014 reported that IGF-1 could be a valuable indicator in predicting pubertal age. A marked positive correlation was demonstrated in IGF-1 levels from the prepubertal to the pubertal stage, but only a moderate negative correlation existed between IGF-1 from the pubertal to postpubertal stage [26]. In this study, we noticed a statistically significant sex variation in the expression of IGF-1 during pubertal onset, where males demonstrated greater IGF-1 levels than females (*p* = 0.03). Female subjects exhibited peak salivary IGF-1 levels compared to males during the stage of puberty and pubertal deceleration, followed by a steep decline in the level of growth completion. These findings correlate to some extent with the results of the study by Ishaq et al., where the peak value in females was exhibited around cervical vertebral staging 4 (CS4) [1]. However, these observations were not in accordance with the reports stated by Juul et al., where increased IGF-1 levels and maximum growth in height were observed one year earlier in girls than boys [31]. The potential sex difference in our study may be due to the unmatched gender sample size and inclusion of subjects with local factors that favor periodontal inflammation. 

Rajpathak et al. stated that low IGF-I and IGFBP-3 levels were associated with systemic markers of inflammation, including high C-reactive protein and interleukin-6 [32]. Additionally, genetic and environmental influences on sex hormones could affect the IGF-dependent action of growth hormones, resulting in variations in IGF-1 levels [27,33]. Therefore, further longitudinal studies were warranted after controlling for confounding factors that could affect the precise estimation. The sex steroid testosterone stimulates growth hormone (GH) through IGF-1 through its direct action on the hypothalamic region, and estrogen inhibits GH-regulated endocrine function in the liver through its indirect action on the pituitary gland. In males, both androgens and estrogens stimulate periosteal bone expansion during puberty, resulting in cortical bone growth [34,35].

Some studies reported that peak statural growth occurs one year later than mandibular growth, and both sexes showed a similar percentage of growth completion at each stage. Therefore, IGF-1 levels were increased in males and females at the same skeletal age [36]. Further investigations have suggested that weight gain results in increased IGF-1 levels in the circulation. In contrast, daily energy expenditure by physical activity would result in a decrease [37,38]. Therefore, sedentary lifestyles among adult female age groups in this study could account for the increase in salivary IGF-1 compared with males. 

There is scarce literature related to salivary IGF-1 levels and skeletal maturity. The results of this study are consistent to some extent with the findings of the study by Nayak S et al. (2014), where salivary IGF-1 levels were lowest at the initial skeletal stages of quantitative cervical vertebral maturation (QCVM) and increased to the highest level at the high-velocity stage [39]. 

IGFBP-3 is the most common IGF-1 binding protein in the blood. Despite its protective impact by controlling the mitogenic action of IGF-1 on cell growth, IGFBP-3 gives an accurate estimation of available freely active IGF-1 present in biological fluids. IGFBPs also function independently of IGF-1 by directly associating with various extracellular and cell surface molecules on mineralized tissues, exhibiting their role in bone modeling function. After performing the IGF generation assay, Blum et al. found IGFBP-3 to be a more accurate discriminator of GH-dependent parameters than IGF-1 [40]. A Study by Toia et al. in 2005 showed detectable changes in salivary IGFBP-3 levels in response to fixed orthodontic force application [16]. Thus, our study aimed to determine the IGFBP-3 levels in saliva to predict growth maturity.

Salivary estimation of IGFBP-3 in this study demonstrated a high degree of significant variation among skeletal stages in both males (*p* = 0.007) and females (*p* = 0.003). The levels of IGFBP3 were found to be accelerated from the prepubertal to peak pubertal stage in male patients, followed by a gradual decline in pubertal deceleration. In this study, we also observed an increase in IGFBP-3 levels in males during the growth completion stage. The levels of IGFBP3 were found to be accelerated during the peak pubertal stage for female patients, followed by a gradual decline from pubertal deceleration (4.06 ng/mL) to growth completion (3.36 ng/mL). Intragroup variation was statistically significant in the prepubertal stage and growth completion stage (*p* < 0.05), as males demonstrated comparatively higher values of IGFBP-3 than females. 

These results were partially in accordance with the study performed by Juul A et al. in 1995, where they found that serum IGFBP-3 levels peaked during pubertal age and that girls experienced one-year peak values earlier than boys. Independent of IGF-1 values, IGFBP-3 values were subject to change depending on the patient’s age, sex, height, body mass index, and pubertal maturation. Therefore, an estimate of the molar ratio of IGF-1/IGFBP-3 would reflect the actual increase in free biologically active IGF-1 [31]. 

The gender differences observed in this study agreed with the study by Lofqvist C et al. in 2005, where they observed that the level of serum IGFBP-3 had a positive age effect in boys, but the level was constant for girls around midpubertal growth. IGFBP-3 values were higher for both boys and girls in midpuberty, followed by a decline as age advanced [41]. Evidence-based studies suggest a varied concentration of IGFBP-3 due to certain nutritional factors and IGFBP-3 promoter polymorphisms affecting the bioavailability of freely active IGF-1 in circulation [33,42].

Despite the influence of sex steroids, environmental and genetic factors on IGF-1dependent growth hormone action, the molar ratio of IGF-1 to IGFBP-3 remained within the normal range regarding age and pubertal stage. Hence, in this study, we determined the relationship of the molar ratio of IGF-1 to IGFBP-3 with different skeletal stages. Statistical analysis of this study by Spearman’s correlation coefficient demonstrated a strong positive association (r = 0.98, *p* < 0.01) between the salivary IGF-1/IGFBP-3 ratio and different stages of skeletal maturity. Significant sex variation was not observed for the IGF-1/IGFBP-3 ratio. 

A study conducted by Kanbur NÖ et al. in 2005 correlated pubertal development characterized according to Tanner’s criteria with the bone formation biomarkers osteocalcin and bone alkaline phosphatase in serum. They stated that the maximum increase in the serum IGF-1/IGFBP-3 molar ratio determines the timing of pubertal growth spurt, which was consistent with the increased bone formation rate during puberty. However, IGF-1 increases with advancing pubertal development, and estimation of the pubertal peak based on IGF-1 alone, especially in girls, was not predictable as IGF-1 levels remained elevated even after the pubertal peak. Therefore, the IGF-1/IGFBP-3 molar ratio may serve as more sensitive yet reliable method of determining growth spurt compared to an estimation of IGF-1 alone [43]. On assessing the results of our study, salivary IGFBP-3 and IGF-1 to IGFBP-3 ratio can serve as a potential biochemical indicator of an adolescent growth spurt, in addition to IGF-1. 

Saliva samples being easy to procure and less invasive can pose a high risk of cross-contamination. A systematic review conducted by Shirazi S in 2021 stated the significance of identifying severe acute respiratory syndrome coronavirus 2 [SARS-CoV-2] among asymptomatic individuals as a part of infection control protocol in the dental environment. Therefore, in the future, COVID-19 point-of-care testing options may be considered as an additional screening tool to detect undetected asymptomatic carriers of SARS-CoV-2 [18,19].

### Strengths and Limitations

The findings of the study support the fact that quantitative assessment of the levels of salivary IGF-1 and IGFBP-3 and their molar ratio can play a significant role in monitoring bone maturity and aid as a clinical diagnostic marker in prioritizing the peak growth period to achieve favorable outcomes during orthodontic intervention. However, certain limitations, such as unmatched sample size related to gender, confounding for the presence of systemic conditions related to hormonal variation such as menstruation, and evaluation of statural parameters could have affected the expression of these biomarkers in the saliva. Despite decreased expression of IGF-1 and IGFBP-3 in saliva compared to serum, saliva could still serve as a non-invasive tool to assess growth maturity. Further longitudinal studies with larger sample sizes are warranted to estimate reference ranges for salivary IGF-1, IGFBP-3, and the molar ratio after adjustment for confounding factors for monitoring and predicting mandibular peak growth spurt in craniofacial orthopedics.

## 5. Conclusions

Since saliva is a non-invasive biological fluid, it can serve as a “mirror of the body” in predicting and monitoring biomarkers related to bone maturation. Under the limitations of this cross-sectional study, we could conclude that salivary IGF-1, salivary IGFBP-3, and salivary IGF/IGFBP-3 ratio serve as potential biochemical markers for assessing pubertal growth spurt and for predicting the completion of skeletal maturity.

## Figures and Tables

**Figure 1 ijerph-19-03723-f001:**
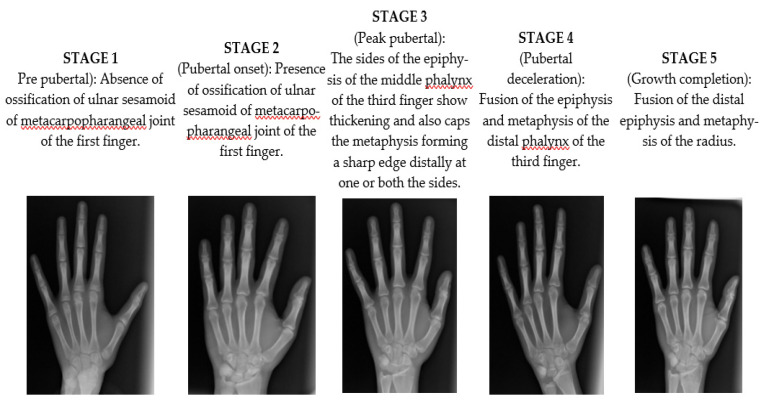
The hand-wrist radiographs depict different skeletal maturity stages.

**Figure 2 ijerph-19-03723-f002:**
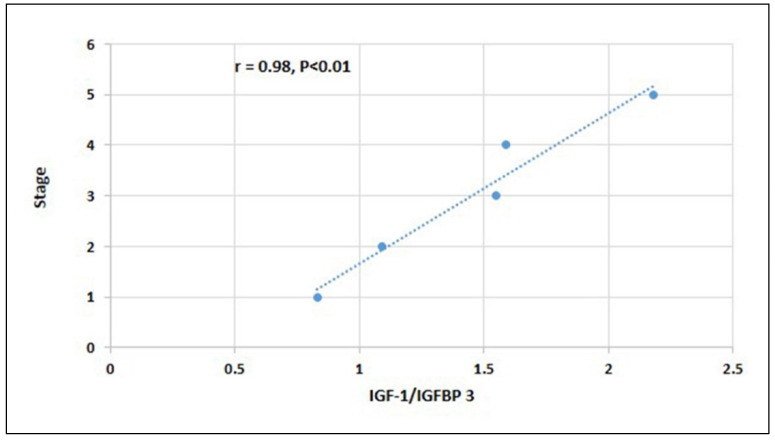
Scatter diagram showing the correlation between IGF/IGFBP3 and different stages of skeletal development as assessed by the hand-wrist radiograph.

**Table 1 ijerph-19-03723-t001:** Frequency and percentage distribution of patients based on gender and various study parameters.

Variable	Class	Male	Female	Total
Age (years)	Less than 14	7 (20.6)	21 (37.5)	28 (31.1)
	14 to 20	13 (38.2)	22 (39.3)	35 (38.9)
	Greater than 20	14 (41.2)	13 (23.2)	27 (30.0)
Location	Urban	9 (26.5)	21 (37.5)	30 (33.3)
	Rural	25 (73.5)	35 (62.5)	60 (66.7)
Family history of periodontal disease	Yes	6 (17.6)	11 (19.6)	17 (18.9)
	No	28 (82.4)	45 (80.4)	73 (81.1)
Frequency of brushing	Once	22 (64.7)	31 (55.4)	53 (58.9)
	Twice	7 (20.6)	25 (44.6)	32 (35.6)
	Never or occ	5 (14.7)	0 (0.0)	5 (5.5)
Previous dental visits	Once a year	9 (26.5)	25 (44.6)	34 (37.8)
	Twice a year	5 (14.7)	15 (26.8)	20 (22.2)
	Occasionally	12 (35.3)	12 (21.4)	24 (26.7)
	Never	8 (23.5)	4 (7.2)	12 (13.3)
Simplified oral hygiene index (OHI-S)	Good	18 (52.9)	34 (60.7)	52 (57.8)
	Fair	11 (32.4)	19 (33.9)	30 (33.3)
	Poor	5 (14.7)	3 (5.4)	8 (8.9)
Bleeding on probing % of sites	No BOP	15 (44.1)	32 (57.1)	47 (52.2)
	<10%	4 (11.8)	6 (10.7)	10 (11.1)
	10–30%	5 (14.7)	9 (16.1)	14 (15.6)
	>30%	10 (29.4)	9 (16.1)	19 (21.1)
Probing pocket depth	Normal	26 (76.5)	46 (82.1)	72 (80.0)
	4–5 mm	8 (73.5)	8 (14.3)	16 (17.8)
	>5 mm	0 (0.0)	2 (3.6)	2 (2.2)
Community periodontal index (CPI)	0	15 (44.1)	31 (55.4)	46 (51.1)
	1	6 (17.6)	6 (10.7)	12 (13.3)
	2	9 (26.5)	12 (21.4)	21 (23.3)
	3	4 (11.8)	5 (8.9)	9 (10.0)
	4	0 (0.0)	2 (3.6)	2 (2.3)
Clinical attachment loss (CAL)	No CAL	29 (85.3)	49 (87.4)	78 (86.7)
	1–2 mm	5 (14.7)	2 (3.6)	7 (7.8)
	3–4 mm	0 (0.0)	3 (5.4)	3 (3.3)
	5 mm and more	0 (0.0)	2 (3.6)	2 (2.2)

**Table 2 ijerph-19-03723-t002:** Comparison of skeletal stages as determined by the hand-wrist radiograph based on IGF-1 and IGFBP-3 in male and female patients (Kruskal–Wallis test and Mann–Whitney test).

Stages	IGF-1			IGFBP-3		
	Male	Female	MW (*p* Value)	Male	Female	MW (*p* Value)
1	0.93 (1.68) ^a^	0.57 (2.48) ^a^	0.156 NS	3.74 (0.34) ^a^	3.23 (1.86) ^a^	0.001 **
2	2.52 (0.38) ^a^	0.87 (1.24) ^abc^	0.033 *	3.96 (0.43) ^ab^	3.31 (2.07) ^a^	0.155 NS
3	2.58 (1.10) ^a^	2.22 (1.50) ^b^	0.770 NS	4.29 (0.21) ^b^	4.31 (0.83) ^b^	0.767 NS
4	1.67 (2.48) ^a^	2.32 (1.80) ^bc^	0.518 NS	3.60 (0.85) ^a^	4.06 (1.57) ^ab^	0.253 NS
5	1.09 (1.75) ^a^	1.47 (1.75) ^c^	0.503 NS	4.25 (0.70) ^b^	3.36 (2.13) ^a^	0.000 **
(KW) *p* value	0.231 NS	0.004 **		0.007 **	0.003 **	

(KW: Kruskal–Wallis test, MW: Mann–Whitney test) * denotes statistically significant result, **: significant at 1% level (*p* < 0.01), NS: not significant (*p* > 0.05), Data are represented as median (range). medians with different superscripted alphabets shows pairwise significance of stages by Mann–Whitney test. IGF-1: insulin growth factor-1, IGFBP-3: insulin growth factor binding protein-3. ^a^, ^b^ and ^c^ are alphabets used to denote pairwise comparison. When a common superscripted alphabet occurs at two different group for medians, then it means these groups do not differ significantly.

## Data Availability

Data supporting reported results can be presented on request.

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
