# Peer review of "Correlation between Salivary Levels of IGF-1, IGFBP-3, IGF-1/IGFBP3 Ratio with Skeletal Maturity Using Hand-Wrist Radiographs"

_ijerph, 2022, doi:10.3390/ijerph19063723_

Round 1
Reviewer 1 Report
The research design and topic is interesting for the readers. Overall, it has been presented nicely. However, there are few comments which will help the readers to grasp the topic. I suggest to use different shapes for the graphs in figures 2 and 3. If someone prints in black and white then its difficult to distinguish. Please use (-x-) with the circled one instead of orange and blue. In the discussion section, please add the strengths and limitations of the current study.Author Response
we appreciate your valuable suggestions.
the response to the comments is attached along in a word file.

Reviewer 2 Report
This is a valuable study that investigates the compares salivary levels of IGF-1, IGFBP-3, and IGF-1/IGFBP-3 ratio at different skeletal maturity stages based on skeletal maturity indicators in hand-wrist radiographs. The topic investigated is certainly of interest, however, the following points are to be addressed:
The description of tests in the statistical analysis seems wrong. The Kruskal-Wallis test followed by Bonferroni was used to compare the maturation stages, while the Mann-Whitney test was used to compare the sex categories.
Please, indicate in the tables which summary measures are presented. Are the mean and standard deviation or median and interquartile deviation shown in table 2? If a nonparametric test was used, it is indicated to present the median and interquartile deviation measurements.
Figures 2 and 3 are not unnecessary, they just repeat results that can be seen in Table 2.
The correlation analysis of IGF/IGFBP3 and different stages of skeletal development seems wrong (Figure 4). The most correct analysis would be performed with the calculation of the correlation coefficient between the stage of skeletal development and the IGF-1/IGFBP-3 ratio in the entire sample, and not only between the means by stage group.
Figure 5 seems to deviate from the objective of the study. I recommend removing it.
(Page 9, lines 286-287). The following passage is misinterpreted. “The intergroup correlation between salivary IGF-1 levels and skeletal maturity among females was statistically significant (P=0.004).” In fact, there were differences in salivary IGF-1 level according to the skeletal stage in the female group.
(Page 9, line 288-289). “The Pearson correlation coefficient estimated demonstrated a strong positive correlation (r= 0.6) from stage 1 to stage 4”.? Was this analysis performed?
Author Response
thank you for the valuable suggestions.
the response to the comments is attached along in a word file.

Reviewer 3 Report
The goal of this study was to explore whether minimally invasive salivary estimation of biomarkers Insulin like growth factor (IGF-1) and Insulin like growth factor binding protein-3 (IGFBP-3) could be used to estimate skeletal maturity for clinical convenience, especially in children and adolescent age groups. This is a timely and generally well-written paper. There are a number of questions that arise in the reading of this paper that are reported below.
1/Page 3 line 104. Please clarify this sentence.
2/Lines 117-153, I suggested introducing subheadings, e.g.:
- Lines 117 to 121: Ethics
- Lines 122 to 139: Population
- Lines 140 to 153: Data collection
3/In the discussion section, you should discuss that the results are limited to patients without comorbidity.
Author Response
Thank you for the valuable suggestions.
the response to the comments is been attached as a word file.

Reviewer 4 Report
- Language needs native editing
- Intro is very long. Remove lines 57-67, not necessary
- Move lines 94-103 to discussion
- Line 107-115 say same thing in different ways. Merge and summarize
- How optical density from ELISA were normalized?
- Provide explanation for abbreviations used under each tables
- Clarify different superscripted alphabets in tables
- What are numbers in parenthesis in tables?
- What is MU in table2?
- Why median values were used in figure 2 and 3, not mean and SD
- In conclusion, mention “under the limitations of this study”
Author Response
thank you for the valuable suggestions.
the response to the comments and the certificate for the language edition has been attached along.

Round 2
Reviewer 2 Report
The authors have addressed all the points raised. I have no more comments.
Author Response
thank you for reviewing the manuscript.

Reviewer 4 Report
- In section 2.3, data collection, since the authors mention COVID-19 and the fact that this study was conducted during pandemic and using saliva, I suggest adding some details on screening procedures. The bellow references from ADA and JCM are suggested. Please utilize and refer to these articles in this section
Testing for COVID-19 in dental offices: Mechanism of action, application, and interpretation of laboratory and point-of-care screening tests. The Journal of the American Dental Association 152 (7), 514-525. e8
Characteristics and detection rate of SARS-CoV-2 in alternative sites and specimens pertaining to dental practice: an evidence summary. Journal of Clinical Medicine 10 (6), 1158.
- Regarding the ELISA test, my concern is still not answered. In line 152 the authors mention that the samples were centrifuged “for 10 min to remove proteins”. This is not correct. ELISA tests measure protein, so you cannot eliminate protein. Samples should be centrifuged to remove particles. Another concern is that the authors mentioned they used equal volumes of saliva for the test. This is also wrong. Same amounts of protein should be used for ELISA because same volumes of substrate may have different protein content and the results will be inaccurate due to unequal amount of starting substrate. Total protein concentration should have been quantified using a total protein assay and same amounts of protein should have been used in ELISA.
Author Response
Thank you for your valuable comments.
